# Radiocesium transfer rates among pigs fed haylage contaminated with low levels of cesium at two differentiation stages

**Chunxiang Piao[1,2], Min Ma[1], James K. Chambers[3], Kazuyuki Uchida[3], Masanori Ikeda[1], Natsuko I. Kobayashi[4], Atsushi Hirose[4], Keitaro Tanoi[4], Masayoshi Kuwahara[5], Junyou Li[1]***

**1** Animal Resource Science Center, Graduate School of Agricultural and Life Sciences, The University of Tokyo, Kasama, Japan, **2** Agricultural Resources and Environment, Faculty of Agricultural, Yanbian University, Jilin, China, **3** Laboratory of Veterinary Pathology, Graduate School of Agricultural and Life Sciences, The University of Tokyo, Tokyo, Japan, **4** Isotope Facility for Agricultural Education and Research, Graduate School of Agricultural and Life Sciences, The University of Tokyo, Tokyo, Japan, **5** Veterinary Pathophysiology and Animal Health, Graduate School of Agricultural and Life Sciences, The University of Tokyo, Tokyo, Japan

* ajunyou@mail.ecc.u-tokyo.ac.jp

**Data Availability Statement:** All relevant data are within the paper and figures.

**Funding:** This study was funded by a Japan Society for the Promotion of Science KAKENHI

## Abstract

The objective of this study was to determine the radiocesium transfer rates of pigs fed haylage contaminated with low levels of cesium at different growth stages. We measured the body weight of juvenile and adult pigs during the treatment period to confirm their health status. We also performed pig blood hematologic and biochemical analyses at both growth stages. To our knowledge, this is the first study to report pig radiocesium transfer coefficient rates after 1 month of chronic oral treatment, which is the period assumed to be required for body equilibrium under a diet of radiocesium-contaminated food. The results showed higher radiocesium retention rates in the kidneys, liver, spleen, genitals, psoas major, bladder, thyroid, and urine than in the blood and bone (tibia and femur) of pigs at both growth stages. The radiocesium retention levels were generally higher in juvenile pigs than in adult pigs, with the highest transfer coefficient ratio in the kidneys (16.2%).

## Introduction

On March 11, 2011, a magnitude 9.1 earthquake occurred off the Pacific coast of Tohoku, Japan. This earthquake triggered a powerful tsunami that destroyed local villages and took a devastating toll on human life. The earthquake and subsequent tsunami also led to the Fukushima Daiichi nuclear power plant disaster, in which a nuclear reactor meltdown resulted in the release of large amounts of radioactive material, contaminating plants and animals in a wide surrounding area. The Animal Resource Science Center (ARSC) of the University of Tokyo (Tokyo, Japan) is located 140 km southwest of the Fukushima Daiichi nuclear power plant. Contamination of the ARSC pastures has been confirmed [1]. However, Italian ryegrass that had been seeded in September 2010 was harvested in May 2011 as usual. The raw grass was

Grant-in-Aid for Scientific Research (C) (no. JP25517004) and by the Livestock Promotion Agency of Japan Racing Association. The funding of "Japan Racing Association", the project leader is our colleague, same laboratory. I am a member of this project. The present research used pocH-100iV Diff hematology analyzer (Sysmex, Hyogo, Japan) for hematological parameters analyze, and using a DRI-CHEM 3500s (Fujifilm, Japan) did blood biochemistry analyze. Both of which was purchased by above grant. But present research mainly done by the grant of "Japan Society for the Promotion of Science KAKENHI Grant-in-Aid for Scientific Research (C) (no. JP25517004)". I am the leader of this project. Both of the funders above had no role in study design, data collection and analysis, decision to publish, or preparation of the manuscript.

**Competing interests:** The funding of "Japan Racing Association", the project leader is our colleague, same laboratory. I am a member of this project. The present research used pocH-100iV Diff hematology analyzer (Sysmex, Hyogo, Japan) for hematological parameters analyze, and using a DRI-CHEM 3500s (Fujifilm, Japan) did blood biochemistry analyze. Both of which was purchased by above grant. But present research mainly done by the grant of "Japan Society for the Promotion of Science KAKENHI Grant-in-Aid for Scientific Research (C) (no. JP25517004)". I am the leader of this project. The corresponding author, on behalf of all authors declare no competing interests associated with this manuscript. This does not alter our adherence to PLOS ONE policies on sharing data and materials.

dried for several days and then packed in plastic film. The grass was allowed to ferment for 1 month for conversion into animal feed as haylage. The haylage was confirmed on June 3, 2011, to be contaminated with radioactive nuclides at 1.91–2.18 kBq/kg $^{134}$C and 2.03–2.20 kBq/kg $^{137}$C; however, $^{131}$I was not detected [2]). Within 2 months after the Fukushima Daiichi nuclear power plant accident, this contaminated haylage was used to study radiocesium transfer ratios from feed to cow milk. Cesium (Cs) is a naturally occurring element found at low concentrations in rocks, soil, and dust. Natural Cs is present in the environment in only one stable form, as the isotope $^{133}$Cs. Radioactive forms of Cs are unstable and are eventually transformed into other, more stable elements through radioactive decay. The two most important radioactive Cs isotopes are $^{134}$Cs and $^{137}$Cs. Each atom of $^{134}$Cs changes into either xenon-134 ($^{134}$Xe) or barium-134 ($^{134}$Ba), neither of which is radioactive, whereas $^{137}$Cs atoms decay to barium-137 ($^{137}$Ba), which is also not radioactive. As $^{134}$Cs and $^{137}$Cs decay, beta particles and gamma radiation are given off. The half-life, or time required for half of the isotope to give off radiation and transform into a different element, of $^{134}$Cs is about 2 years, whereas that of $^{137}$Cs is about 30 years [3].

In the present study, we examined the health status of 26 pigs that had been raised 17 km northwest of the Fukushima Daiichi nuclear power plant. These pigs were rescued about 110 days after the disaster, and subsequently maintained for over 3 months on an uncontaminated feedlot at the ARSC, after which radiocesium contamination was detected in multiple tissue samples (ovary, testis, spleen, liver, kidney, psoas major, urine, and blood) [1]. The pig farm owners confirmed that the pigs were not fed radiation-contaminated concentrate and that they had access to uncontaminated groundwater while they lived within the radiation alert area. However, radiocesium was detected in pig organs 9 months after the nuclear disaster. Thus, removal of radiocesium from the pig body requires a relatively long period. A previous study reported that the decrease in whole-body activity among voles fed regularly with $^{134}$Cs dropped to below the minimum detectable level after 21 days [4]; however, our own previous findings [1] did not confirm those of this early report.

Our previous hematological and biochemical index analyses showed different plasma serum chemistry values between pigs rescued from the nuclear disaster area and normal pigs fed at the ARSC [1]. We also detected reproductive dysfunction among rescued sows fed uncontaminated concentrate and groundwater at the ARSC. In the present study, we examined radiocesium transfer rates in organ tissues of pigs fed with haylage contaminated with low levels of Cs for 1 month at different growth stages. We measured radiocesium levels in all organ tissue samples and calculated hematological and biochemical indices to confirm the health status of the pigs. Because pigs share many physiological similarities with humans and offer breeding and handling advantages compared to non-human primates, they are model organisms for preclinical experimentation in the fields of reproductive physiology, xenotransplantation, and neurotransplantation. Therefore, the results of the present study may have broader research applications for humans and other animal species.

## Materials and methods

All experiments were approved by the Animal Care and Use Committee of the Graduate School of Agricultural and Life Sciences at the University of Tokyo (Approval ID: P15-061).

In this study, we used 10 Middle Yorkshire × Duroc juvenile pigs, including castrated males and females. Each treatment group contained five pigs (mean body weight: 57.9 ± 10.9 kg). The control group also consisted of five pigs (mean body weight: 54.8 ± 12.9 kg); this group was used only for comparing body weight and hematological and biochemical indices.

We also used four adult Middle Yorkshire pigs (all male; mean body weight: 244.3 ± 11.0 kg). There was no adult pig control group; therefore, the hematological and biochemical indices were compared among pigs of the same age.

## Radiocesium treatment

The pigs were fed radiocesium-contaminated haylage (620–709 Bq/kg Cs) consisting of Italian ryegrass grown from September 2010 to May 2011 in an ARSC field. This study was conducted 2.5 years after the 2011 Fukushima Daiichi nuclear power plant disaster; therefore, radioactivity levels have declined to some extent. The haylage was cut to a length of 1.3 cm using a power feed cutter (Cowa Cutter Inc., Mishima, Shizuoka, Japan) and mixed with concentrate (0.7 kg contaminated haylage + 0.7–1.5 kg concentrate). The control group feed consisted of uncontaminated Italian ryegrass (harvested 2013) haylage cut to a length of 1.3 cm and mixed with concentrate (0.7 kg uncontaminated haylage + 0.7–1.5 kg concentrate).

## Feed and treatment period

Contaminated haylage (0.7 kg) was mixed with feed concentrate daily for 1 month, at which point the highest cumulative radiocesium level was expected to be reached. After 1 month of treatment, all pigs were provided only uncontaminated concentrate (1.5–2.2 kg/day) until the end of the study period.

## Radiocesium transfer coefficients

The general formula for transfer coefficients of nuclide transfer from feed to animal products is defined as [5, 6]:

Transfer coefficient (%) = Concentration in meat or organ (Bq/kg)/Daily nuclide intake (Bq/animal/day)×100

The daily nuclide intake (Bq/day) is calculated as radiocesium concentration in feed (Bq/kg) × feed intake (kg/day). A longer period is required for pig meat or organs to reach equilibrium. In the present study, pigs were fed contaminated feed for 1 month to reach equilibrium (or the highest cumulative radiocesium level).

The feed concentrate was purchased from Chubu Shiryo Inc. (Nagoya, Japan); its composition was 75.5% total digestible nutrients (TDN), 15.5% crude protein (CP), 3.5% ether extract (EE), 6.0% crude fiber (CF), 0.75% calcium (Ca), and 0.7% phosphorous (P).

## Blood sample collection

Blood samples were collected from the jugular vein using an 18-gauge needle (Venoject II; Terumo, Tokyo, Japan) into 5-mL collection tubes containing EDTA-Na. The hematological parameters were analyzed using a pocH-100iV Diff hematology analyzer (Sysmex, Hyogo, Japan), and the blood tubes were then centrifuged for 20 min at 3,000 rpm and 4°C to obtain plasma. To determine blood biochemistry parameters, the plasma was analyzed using DRI-CHEM slides (Fujifilm, Tokyo, Japan) and a DRI-CHEM 3500s blood biochemistry analyzer (Fujifilm).

Blood samples were collected from juvenile pigs six times, at 1 day before treatment (Day 0), 2 weeks after the start of treatment, at the end of treatment (1 month after the start of treatment), and then at 1, 3, and 5 months after the end of treatment. Blood samples were collected from adult pigs on the same schedule.

### Tissue and organ sample selection

Tissue and organ samples were collected from the liver, spleen, kidneys, genitals, psoas major, bladder, thyroid, urine, thymus, tibia (left hind leg), and femur (left hind leg) of pigs following euthanization with Ravonal (Nipro ES Pharma Co. Ltd., Osaka, Japan) and exsanguination as described previously [7]. These methods were in accordance with the approved guidelines.

Organ and tissue samples were collected from juvenile pigs five times, at 1 month after the start of treatment, and then at 1, 3, 7, and 8 months after the end of treatment. Organ and tissue samples were collected from adult pigs four times, at 1 month after the start of treatment, and then at 1, 4, and 6 months after the end of treatment.

### Measurements and analyses

Body weight was measured using a digital balance (MW100-2; Nakajima Seisakusho Co., Ltd., Nagano, Japan).

Hematological analyses were conducted immediately using an automated pocH-100iV Diff hematology analyzer (Sysmex Corp., Kobe, Japan). The hematologic indices included white blood cell (WBC) and red blood cell (RBC) counts, hemoglobin (HGB), hematocrit (HCT), mean corpuscular volume (MCV), mean corpuscular hemoglobin (MCH), mean corpuscular hemoglobin concentration (MCHC), platelet (PLT), WBC/small cell ratio (W-SCR), WBC/mid-cell ratio (W-MCR), WBC/large cell ratio (W-LCR), small white cell count (W-SCC), middle white cell count (W-MCC), and large white cell count (W-LCC).

Biochemical analyses were conducted immediately using an automatic dry-chemistry analyzer (DRI-CHEM 3500s; Fujifilm). The biochemical indices included total protein (TP-P III), albumin (ALB-P), total bilirubin (TBIL-P III), glutamic oxaloacetic transaminase/aspartate aminotransferase (GOT/AST-P III), glutamic pyruvic transaminase/alanine (GPT/ALT-P III), alkaline phosphatase (ALP-P III), gamma glutamyl transferase (GGT-P III), γ-glutamyl transpeptidase (γ-GTP), low-density lipoprotein (LDH-P III), leukocyte alkaline phosphatase (LAP-P), creatine phosphokinase (CPK-P III), amylase (AMYL-P III), ammonia (NH3-P II), total cholesterol (TCHO-P III), high-density lipoprotein (HDL-C-P III), triglyceride (TG-P III), urine (UA-P III), blood urea nitrogen (BUN-P III), creatinine (CRE-P III), glucose (GLU-P III), calcium (Ca-P III), IP-P, and magnesium (Mg-P III).

Radiocesium levels were determined as $^{134}$Cs and $^{137}$Cs counts at the Isotope Facility for Agricultural Education and Research at the University of Tokyo (Tokyo, Japan).

In the present study, we fed haylage contaminated with low levels of radiocesium to pigs over a 1-month period. This diet met the nutrition requirements for swine growth and maintenance according to the National Research Council. Juvenile pigs showed good growth and body weight gain, and adult pigs maintained good body condition throughout the experimental period. No health problems or illnesses were detected.

## Results

### Body weight

In juvenile pigs, body weight gain was normal in both the treatment and control groups, with no difference between groups (Table 1). Adult pigs were found to be in good health, with normal body weights (Table 2).

### Hematology and biochemistry

Hematological and biochemical indices for juvenile pigs in the treatment and control groups are shown in Tables 3 and 4, while those for adult pigs are shown in Tables 5 and 6. The WBC

**Table 1. The bodyweight changes of growing pigs during the treatment periods.**

|  | Day0 (kg) | 1month (kg) | 2months (kg) | 4months (kg) | 5months (kg) | 6months (kg) |
|---|---|---|---|---|---|---|
| Con | 51.5 | 65.5 | 90.4 | 99.3 | 133.7 | 140.8 |
| Tre | 57.9 | 70.3 | 86.3 | 103.0 | 134.4 | 156.0 |

Day0; day of the experiment beginning.

1month: one month age from the experiment beginning and et al.

counts increased during the first month and then decreased. No similar trends were observed for RBCs or other hematological indices during the experimental period.

Few significant changes in the biochemical indices were detected throughout the study period in juvenile and adult pigs (Tables 4 and 6, respectively). ALP-P III differed greatly between the two growth stages; however, no medical condition was recognized. In adult pigs, the WBC counts were lower at 4 months after the end of treatment, but recovered at 6 months (Table 5) and no clinical symptoms associated with low WBCs were observed. After the first week of treatment, TP-P III and ALB-P decreased and NH3-P II increased in adult pigs.

## Radiocesium levels in tissues and organs

In this study, radiocesium radioactivity was determined as total concentrations of $^{134}$Cs and $^{137}$Cs. Radiocesium levels in tissue and organ samples from juvenile pigs are shown in Fig 1. After 1 month of treatment, most samples showed high radiocesium levels, except for blood and bone samples. No radioactivity data were collected for the genitals of juvenile pigs, since the males were castrated at nursing. The highest radiocesium levels in the samples from juvenile pigs were found in the kidneys (105 Bq/kg), whereas the lowest were found in bone (tibia and femur) and were undetectable until 8 months after the end of treatment.

The radiocesium levels in the samples from adult pigs are shown in Fig 2. After 1 month of treatment, radiocesium contamination was detected in most organs, with the highest levels in the liver (44.8 Bq/kg) and kidneys (43.9 Bq/kg), and no detectable contamination in the tibia and femur. These results are consistent with those for the tissue and organ samples from the juvenile pigs. However, a comparison of Figs 1 and 2 shows greater radiocesium accumulation in juvenile pigs than in adult pigs.

## Radiocesium transfer coefficients

In juvenile pigs, the radiocesium transfer coefficient ratios were highest in the kidneys (16.2%) (Table 7). The transfer coefficient ratios were lower in blood and bone than in all other organs. The transfer coefficient ratios of juvenile pigs were roughly double those of adult pigs.

## Discussion

Among juvenile pigs, ALP-P III was significantly different between the control and treatment groups; however, the highest ALP-P III value was a normal level [8] and no medical conditions

**Table 2. The bodyweight changes of adult pigs during the treatment periods.**

| Day0 (kg) | 1month (kg) | 2months (kg) | 4months (kg) | 5months (kg) | 6months (kg) |
|---|---|---|---|---|---|
| 240.9 | 243.3 | 239.8 | 239.8 | 233.1 | 238.2 |

Day0; day of the experiment beginning.

1month: one month age from the experiment beginning and et al.

**Table 3. The hematology index of two groups (growing pig).**

| | n | | WBC (10³/μl) | RBC (10⁴/μl) | HGB (g/dL) | HCT (%) | MCV (fL) | MCH (pg) | MCHC (g/dL) | PLT (10⁴/μl) | W-SCR (%) | W-MCR (%) | W-LCR (%) | W-SCC (10²/μl) | W-MCC (10²/μl) | W-LCC (10²/μl) |
|---|---|---|---|---|---|---|---|---|---|---|---|---|---|---|---|---|
| Day0 | 5 | Average(T) ± SE | 177.0 ± 28.7 | 743.0 ± 66.5 | 12.9 ± 1.1 | 45.9 ± 3.9 | 61.8 ± 1.0 | 17.4 ± 0.3 | 28.2 ± 0.1 | 21.0 ± 9.3 | 75.9 ± 1.7 | 7.4 ± 1.5 | 16.6 ± 1.5 | 133.8 ± 19.0 | 13.2 ± 4.1 | 30.0 ± 6.7 |
| | 5 | Average(C) ± SE | 189.2 ± 38.6 | 721.8 ± 53.9 | 12.6 ± 0.8 | 44.9 ± 3.4 | 62.3 ± 2.5 | 17.5 ± 0.8 | 28.1 ± 0.4 | 28.7 ± 8.6 | 73.8 ± 4.6 | 8.2 ± 1.3 | 18.0 ± 3.9 | 138.2 ± 19.3 | 15.8 ± 6.1 | 35.2 ± 14.0 |
| 2 weeks treatment | 5 | Average(T) ± SE | 218.6 ± 38.1 | 716.2 ± 85.1 | 12.7 ± 1.2 | 44.5 ± 4.6 | 62.2 ± 2.1 | 17.8 ± 0.6 | 28.6 ± 0.4 | 28.3 ± 8.2 | 66.4 ± 3.6 | 10.5 ± 1.5 | 23.1 ± 2.9 | 144.8 ± 20.4 | 23.2 ± 5.3 | 50.6 ± 14.6 |
| | 5 | Average(C) ± SE | 198.2 ± 20.4 | 732.8 ± 40.1 | 13.5 ± 0.6 | 47.1 ± 2.9 | 64.5 ± 1.5 | 18.4 ± 0.5 | 28.6 ± 0.7 | 19.6 ± 8.2 | 73.7 ± 2.5 | 8.2 ± 2.0 | 18.1 ± 1.6 | 146.2 ± 17.1 | 16.0 ± 3.5 | 36.0 ± 5.4 |
| 1 month treatment | 5 | Average(T) ± SE | 180.8 ± 30.5 | 790.2 ± 102.0 | 14.3 ± 1.7 | 49.5 ± 6.1 | 62.6 ± 1.8 | 18.1 ± 0.5 | 28.9 ± 0.5 | 25.8 ± 8.4 | 74.5 ± 2.8 | 7.7 ± 0.8 | 17.7 ± 3.0 | 134.4 ± 18.7 | 14.0 ± 3.0 | 32.4 ± 10.4 |
| | 5 | Average(C) ± SE | 167.4 ± 35.3 | 760.2 ± 65.8 | 14.0 ± 1.3 | 48.7 ± 4.3 | 64.1 ± 1.2 | 18.5 ± 0.5 | 28.8 ± 0.5 | 30.7 ± 6.6 | 72.6 ± 5.8 | 8.0 ± 1.3 | 19.4 ± 5.7 | 121.6 ± 28.8 | 13.4 ± 3.2 | 32.4 ± 10.5 |
| 1 month age | 4 | Average(T) ± SE | 143.7 ± 25.6 | 756.3 ± 43.9 | 14.4 ± 0.8 | 48.9 ± 3.0 | 64.7 ± 2.0 | 19.0 ± 0.5 | 29.4 ± 0.3 | 30.5 ± 7.9 | 66.5 ± 10.6 | 8.6 ± 3.4 | 25.0 ± 7.5 | 95.0 ± 18.7 | 12.7 ± 5.9 | 36.0 ± 13.5 |
| | 4 | Average(C) ± SE | 168.8 ± 22.8 | 736.8 ± 28.2 | 14.0 ± 0.7 | 47.5 ± 2.0 | 64.5 ± 1.0 | 18.9 ± 0.3 | 29.4 ± 0.2 | 29.6 ± 5.7 | 63.5 ± 4.0 | 10.2 ± 1.3 | 26.3 ± 4.0 | 107.0 ± 15.7 | 17.3 ± 3.2 | 44.5 ± 9.9 |
| 3 months age | 3 | Average(T) ± SE | 142.0 ± 15.5 | 735.0 ± 52.1 | 14.0 ± 0.7 | 47.8 ± 3.1 | 65.1 ± 3.0 | 19.1 ± 0.7 | 29.4 ± 0.7 | 26.9 ± 4.6 | 60.9 ± 6.1 | 10.0 ± 3.2 | 29.1 ± 3.1 | 87.0 ± 15.6 | 14.0 ± 3.6 | 41.0 ± 5.6 |
| | 4 | Average(C) ± SE | 157.8 ± 42.5 | 822.3 ± 44.3 | 15.8 ± 0.9 | 53.5 ± 2.6 | 65.1 ± 1.9 | 19.2 ± 0.2 | 29.6 ± 0.7 | 17.4 ± 4.2 | 61.8 ± 5.9 | 8.4 ± 1.2 | 29.8 ± 5.0 | 96.0 ± 22.7 | 13.3 ± 4.9 | 48.5 ± 17.5 |
| 5 months age | 1 | Average(T) | 135.0 | 683.0 | 13.2 | 42.8 | 62.7 | 19.3 | 30.8 | 23.9 | 73.2 | 11.4 | 15.4 | 99.0 | 15.0 | 21.0 |
| | 1 | Average(C) | 153.0 | 702.0 | 13.6 | 45.0 | 64.1 | 19.4 | 30.2 | 20.1 | 60.1 | 13.9 | 26.0 | 92.0 | 21.0 | 40.0 |

Day0: The day of begining of treatment

All data are expessed as mean±S.E.

S.E.: standard error

Average(T): fed the cesium contaminated haylage

Average(C): fed the none cesium contaminated haylage

2 week treatment: fed contaminated haylage for 2 week

1 month treatment: fed contaminated haylage for1 month

1 month age: 1 month after from stopped fed contaminated haylage

3 months age: 3 month after from stopped fed contaminated haylage

5 months age: 5 month after from stopped fed contaminated haylage

**Table 4. The biochemial index of two groups (growing pig).**

| | n | | | TP-PIII (g/dl) | ALB-P (g/dl) | TBIL-PIII (mg/dl) | AST-PIII (U/l) | ALT-PIII (U/l) | ALP-PIII (U/l) | GGT-PIII (U/l) | LDH-PIII (U/l) | LAP-P (U/l) | CPK-PIII (U/l) | AMYL-PIII (U/l) | NH3-PI (μg/dl) | TCHO-PIII (mg/dl) | HDL-C-PIII (mg/dl) | TG-PIII (mg/dl) | UA-PIII (mg/dl) | BUN-PIII (mg/dl) | CRE-PIII (mg/dl) | GLU-PIII (mg/dl) | Ca-PIII/IP-P·Mg-PIII (mg/dl)/(mg/dl)/(mg/dl) |
|---|---|---|---|---|---|---|---|---|---|---|---|---|---|---|---|---|---|---|---|---|---|---|
| Day0 | 5 | Average(T) | ± SE | 6.2 ± 0.3 | 5.3 ± 0.7 | 0.1 ± 0.1 | 42.2 ± 9.0 | 64.8 ± 4.9 | 240.0 ± 207.1 | 53.6 ± 20.1 | 678.2 ± 45.4 | 38.4 ± 10.9 | 647.8 ± 198.0 | 1457.0 ± 718.1 | 92.0 ± 24.6 | 80.4 ± 5.5 | 29.2 ± 3.5 | 22.8 ± 11.1 | 0.3 ± 0.0 | 15.7 ± 3.1 | 1.2 ± 0.2 | 98.8 ± 18.9 | 8.7± 4.1 9.0 ± 0.5 1.7 ± 0.4 |
| | 5 | Average(C) | ± SE | 6.5 ± 0.5 | 5.7 ± 0.2 | 0.1 ± 0.0 | 57.8 ± 15.4 | 61.4 ± 7.3 | 32.0 ± 13.4 | 98.8 ± 40.2 | 619.8 ± 30.0 | 27.2 ± 2.9 | 897.0 ± 355.6 | 664.4 ± 179.5 | 95.6 ± 40.3 | 86.6 ± 9.8 | 30.4 ± 4.2 | 40.8 ± 22.6 | 0.3 ± 0.0 | 11.9 ± 2.1 | 1.1 ± 0.2 | 92.0 ± 11.1 | 4.2± 0.1 9.9 ± 0.8 1.2 ± 0.1 |
| 2 weeks treatment | 5 | Average(T) | ± SE | 6.0 ± 0.9 | 4.4 ± 0.8 | 0.1 ± 0.0 | 30.8 ± 6.4 | 71.8 ± 19.4 | 322.2 ± 49.7 | 32.4 ± 7.8 | 525.2 ± 193.9 | 47.4 ± 3.0 | 517.0 ± 330.3 | 1801.8 ± 392.3 | 88.2 ± 29.3 | 82.4 ± 13.8 | 32.6 ± 4.3 | 22.2 ± 13.4 | 0.3 ± 0.0 | 9.4 ± 1.7 | 1.1 ± 0.2 | 99.0 ± 13.9 | 11.9± 0.8 7.9 ± 1.0 2.0 ± 0.1 |
| | 5 | Average(C) | ± SE | 5.4 ± 0.7 | 4.3 ± 0.5 | 0.1 ± 0.0 | 28.2 ± 2.8 | 42.2 ± 5.8 | 309.4 ± 152.6 | 31.2 ± 6.3 | 472.2 ± 58.9 | 46.0 ± 11.5 | 997.4 ± 633.5 | 1698.4 ± 747.8 | 68.4 ± 10.4 | 75.6 ± 15.0 | 34.0 ± 4.7 | 30.0 ± 18.7 | 0.2 ± 0.0 | 13.6 ± 2.2 | 1.0 ± 0.2 | 105.2 ± 8.6 | 9.6± 3.3 7.7 ± 0.7 2.0 ± 0.5 |
| 1 month treatment | 5 | Average(T) | ± SE | 6.7 ± 0.7 | 5.1 ± 0.2 | 0.1 ± 0.1 | 56.8 ± 31.4 | 87.4 ± 17.7 | 3.6 ± 5.3 | 53.2 ± 23.5 | 634.5 ± 46.5 | 28.0 ± 2.6 | 984.8 ± 567.5 | 801.0 ± 377.8 | 106.2 ± 52.4 | 84.4 ± 12.2 | 32.4 ± 3.6 | 14.8 ± 4.9 | 0.3 ± 0.1 | 9.5 ± 1.2 | 1.3 ± 0.2 | 101.4 ± 11.2 | 3.9± 0.1 8.0 ± 1.1 1.4 ± 0.0 |
| | 5 | Average(C) | ± SE | 6.6 ± 0.3 | 5.5 ± 0.3 | 0.1 ± 0.0 | 33.8 ± 13.7 | 40.6 ± 6.9 | 21.4 ± 12.0 | 46.0 ± 13.8 | 462.0 ± 139.2 | 29.0 ± 4.4 | 535.0 ± 42.5 | 655.2 ± 136.2 | 73.6 ± 20.3 | 93.2 ± 12.4 | 41.0 ± 5.3 | 43.6 ± 22.8 | 0.3 ± 0.0 | 13.2 ± 3.3 | 1.2 ± 0.2 | 101.8 ± 7.5 | 3.9± 0.1 8.1 ± 0.4 1.4 ± 0.1 |
| 1 month age | 4 | Average(T) | ± SE | 6.6 ± 0.8 | 5.6 ± 0.1 | 0.1 ± 0.0 | 37.0 ± 7.5 | 35.3 ± 5.7 | 62.7 ± 8.5 | 66.3 ± 40.8 | 484.0 ± 75.9 | 35.3 ± 5.5 | 928.3 ± 615.5 | 837.7 ± 273.4 | 72.0 ± 12.8 | 73.3 ± 16.8 | 33.0 ± 8.9 | 42.7 ± 36.1 | 0.3 ± 0.1 | 15.6 ± 1.6 | 1.2 ± 0.2 | 100.3 ± 9.3 | 4.0± 0.1 6.6 ± 0.3 1.7 ± 0.1 |
| | 4 | Average(C) | ± SE | 6.5 ± 0.3 | 5.6 ± 0.1 | 0.1 ± 0.0 | 35.8 ± 8.5 | 47.3 ± 14.7 | 115.0 ± 28.5 | 56.0 ± 28.6 | 526.5 ± 70.2 | 39.8 ± 4.9 | 766.0 ± 560.8 | 1003.8 ± 298.7 | 64.3 ± 11.6 | 54.3 ± 5.4 | 22.8 ± 4.6 | 38.3 ± 18.6 | 0.3 ± 0.0 | 10.3 ± 0.5 | 1.3 ± 0.3 | 104.3 ± 14.4 | 4.1± 0.1 7.9 ± 1.3 1.6 ± 0.1 |
| 3 months age | 3 | Average(T) | ± SE | 6.8 ± 0.3 | 5.8 ± 0.2 | 0.1 ± 0.0 | 27.7 ± 4.6 | 36.7 ± 6.5 | 163.7 ± 177.0 | 40.7 ± 18.0 | 442.3 ± 43.8 | 40.3 ± 8.4 | 435.3 ± 83.5 | 1364.7 ± 806.1 | 67.0 ± 25.5 | 74.0 ± 5.3 | 29.0 ± 3.6 | 22.0 ± 12.2 | 0.3 ± 0.1 | 16.4 ± 2.3 | 1.2 ± 0.2 | 93.7 ± 25.4 | 6.8± 4.3 7.9 ± 0.5 2.0 ± 0.3 |
| | 4 | Average(C) | ± SE | 7.4 ± 0.3 | 5.8 ± 0.1 | 0.1 ± 0.0 | 27.5 ± 4.2 | 40.0 ± 3.8 | 235.3 ± 135.3 | 31.5 ± 4.0 | 469.0 ± 80.6 | 44.0 ± 12.6 | 590.0 ± 271.5 | 1848.5 ± 612.6 | 79.0 ± 5.7 | 62.5 ± 11.1 | 23.0 ± 3.6 | 24.3 ± 5.1 | 0.3 ± 0.1 | 12.9 ± 2.7 | 1.2 ± 0.3 | 81.3 ± 12.3 | 10.0± 4.2 7.3 ± 0.6 2.2 ± 0.4 |
| 5 months age | 1 | Average(T) | | 6.6 | 5.9 | 0.1 | 75.0 | 64.0 | 64.0 | 113.0 | 459.0 | 33.0 | 661.0 | 632.0 | 81.0 | 82.0 | 36.0 | 35.0 | 0.4 | 17.6 | 1.9 | 112.0 | 4.1 6.2 2.3 |
| | 1 | Average(C) | | 7.5 | 6.0 | 0.1 | 49.0 | 26.0 | 67.0 | 80.0 | 385.0 | 33.0 | 854.0 | 529.0 | 140.0 | 67.0 | 23.0 | 72.0 | 0.4 | 14.4 | 2.1 | 124.0 | 3.9 7.6 1.7 |

Day0: The day of begining of treatment

All data are expessed as mean±S.E.

S.E.: standard error

Average(T): fed the cesium contaminated haylage

Average(C): fed the none cesium contaminated haylage

2 week treatment: fed contaminated haylage for 2 week

1 month treatment: fed contaminated haylage for1 month

1 month age: 1 month after from stopped fed contaminated haylage

3 months age: 3 month after from stopped fed contaminated haylage

5 months age: 5 month after from stopped fed contaminated haylage

**Table 5. The hematology index of adult pig.**

| | n | | WBC (10²/µl) | | | RBC (10⁴/µl) | | | HGB (g/dL) | | | HCT (%) | | | MCV (fL) | | | MCH (pg) | | | MCHC (g/dL) | | | PLT (10⁴/µl) | | | W-SCR (%) | | | W-MCR (%) | | | W-LCR (%) | | | W-SCC (10²/µl) | | | W-MCC (10²/µl) | | | W-LCC (10²/µl) | | |
|---|---|---|---|---|---|---|---|---|---|---|---|---|---|---|---|---|---|---|---|---|---|---|---|---|---|---|---|---|---|---|---|---|---|---|---|---|---|---|---|---|---|---|---|---|
| Day (0) | 4 | Average ± SE | 156.0 | ± | 33.0 | 721.8 | ± | 31.1 | 14.1 | ± | 0.4 | 47.3 | ± | 1.3 | 65.6 | ± | 1.9 | 19.5 | ± | 0.5 | 29.7 | ± | 0.2 | 29.9 | ± | 2.4 | 44.2 | ± | 10.0 | 11.5 | ± | 1.8 | 44.4 | ± | 11.5 | 66.0 | ± | 5.7 | 17.5 | ± | 2.6 | 72.5 | ± | 34.3 |
| 1 week treatment | 4 | Average ± SE | 145.8 | ± | 18.2 | 782.3 | ± | 142.3 | 13.1 | ± | 0.6 | 51.7 | ± | 9.7 | 66.0 | ± | 1.8 | 18.7 | ± | 0.5 | 28.2 | ± | 0.3 | 31.3 | ± | 5.9 | 41.8 | ± | 10.9 | 8.3 | ± | 2.3 | 50.0 | ± | 12.7 | 62.3 | ± | 20.5 | 12.3 | ± | 3.3 | 71.3 | ± | 14.9 |
| 1 month treatment | 4 | Average ± SE | 141.5 | ± | 14.0 | 929.3 | ± | 187.1 | 17.7 | ± | 2.9 | 61.7 | ± | 10.3 | 66.8 | ± | 2.4 | 19.2 | ± | 0.8 | 28.8 | ± | 0.6 | 19.7 | ± | 10.5 | 47.9 | ± | 12.7 | 11.0 | ± | 1.2 | 41.2 | ± | 12.3 | 66.3 | ± | 14.6 | 15.8 | ± | 2.2 | 59.5 | ± | 23.7 |
| 1 month age | 3 | Average ± SE | 89.3 | ± | 31.5 | 1145.7 | ± | 258.0 | 22.2 | ± | 4.8 | 78.6 | ± | 17.2 | 68.8 | ± | 1.1 | 19.4 | ± | 0.3 | 28.3 | ± | 0.0 | 20.8 | ± | 10.8 | 49.8 | ± | 5.5 | 13.5 | ± | 2.1 | 36.7 | ± | 3.5 | 46.3 | ± | 21.5 | 11.7 | ± | 2.4 | 31.3 | ± | 7.7 |
| 4 months age | 2 | Average ± SE | 92.5 | ± | 5.5 | 849.5 | ± | 118.5 | 16.0 | ± | 2.5 | 55.0 | ± | 8.3 | 64.6 | ± | 0.7 | 18.8 | ± | 0.3 | 29.1 | ± | 0.2 | 23.3 | ± | 2.1 | 60.5 | ± | 0.4 | 9.2 | ± | 1.3 | 30.4 | ± | 0.9 | 56.0 | ± | 3.0 | 8.5 | ± | 1.5 | 28.0 | ± | 1.0 |
| 6 months age | 1 | Average ± SE | 148.0 | | | 826.0 | | | 15.2 | | | 53.0 | | | 64.2 | | | 18.4 | | | 28.7 | | | 28.6 | | | 48.7 | | | 15.2 | | | 26.1 | | | 72.0 | | | 22.0 | | | 54.0 | | |

All data are expessed as mean±S.E.

S.E.: standard error

Day0: The day of beginning of treatment

1 week treatment: fed contaminated haylage for 1 week

1 month treatment: fed contaminated haylage for1 month

1 month age: 1 month after from stopped fed contaminated haylage

4 months age: 4 month after from stopped fed contaminated haylage

6 months age: 6 month after from stopped fed contaminated haylage

**Table 6. The hematology index of adult pig.**

| | n | | | TP-PIII (g/dl) | ALB-P (g/dl) | TBIL-PIII (mg/dl) | AST-PIII (U/l) | ALT-PIII (U/l) | ALP-PIII (U/l) | GGT-PIII (U/l) | LDH-PIII (U/l) | LAP-P (U/l) | CPK-PIII (U/l) | AMYL-PIII (U/l) | NH3-PII (µg/dl) | TCHO-PIII (mg/dl) | HDL-C-PIII (mg/dl) | TG-PIII (mg/dl) | UA-PIII (mg/dl) | BUN-PIII (mg/dl) | CRE-PIII (mg/dl) | GLU-PIII (mg/dl) | Ca-PIII P-P Mg-PIII (mg/dl)(mg/dl)(mg/dl) |
|---|---|---|---|---|---|---|---|---|---|---|---|---|---|---|---|---|---|---|---|---|---|---|---|
| Day (0) | 4 | Average ± SE | | 8.8 ± 0.3 | 5.9 ± 0.1 | 0.1 ± 0.0 | 128.8 ± 31.1 | 32.8 ± 8.6 | 25.5 ± 2.3 | 303.8 ± 67.2 | 518.3 ± 157.2 | 44.3 ± 3.3 | 1701.5 ± 517.0 | 511.5 ± 151.0 | 98.0 ± 39.1 | 66.8 ± 10.9 | 20.8 ± 3.9 | 62.5 ± 12.4 | 0.5 ± 0.0 | 12.8 ± 3.0 | 1.6 ± 0.1 | 94.0 ± 6.8 | 4.1± 0.1 6.3 ± 0.5 1.5 ± 0.0 |
| 1 week treatment | 4 | Average ± SE | | 6.4 ± 0.7 | 5.0 ± 0.4 | 0.1 ± 0.0 | 95.3 ± 39.9 | 34.0 ± 6.4 | 16.8 ± 6.6 | 209.3 ± 40.1 | 900.0 ± 0.0 | 14.8 ± 5.1 | 1527.8 ± 449.5 | 389.8 ± 141.6 | 427.8 ± 59.8 | 60.8 ± 16.9 | 18.0 ± 2.4 | 55.5 ± 11.9 | 0.4 ± 0.0 | 11.1 ± 2.4 | 2.3 ± 0.3 | 94.5 ± 5.5 | 4.2± 0.2 5.4 ± 0.8 1.5 ± 0.2 |
| 1 month treatment | 4 | Average ± SE | | 8.0 ± 0.7 | 5.9 ± 0.1 | 0.1 ± 0.0 | 64.0 ± 23.3 | 43.0 ± 6.0 | 19.8 ± 9.0 | 159.0 ± 76.5 | 398.0 ± 87.4 | 34.0 ± 7.4 | 1079.8 ± 331.1 | 633.5 ± 246.6 | 78.8 ± 28.7 | 54.3 ± 12.6 | 19.8 ± 5.4 | 30.0 ± 8.8 | 0.4 ± 0.0 | 9.3 ± 0.8 | 1.7 ± 0.2 | 85.0 ± 22.7 | 4.2± 0.2 5.9 ± 0.2 1.3 ± 0.1 |
| 1 month age | 3 | Average ± SE | | 8.1 ± 0.2 | 5.8 ± 0.2 | 0.1 ± 0.0 | 54.0 ± 5.7 | 30.3 ± 4.8 | 13.3 ± 2.9 | 104.0 ± 75.7 | 358.0 ± 74.3 | 34.0 ± 1.4 | 1415.7 ± 424.9 | 1006.0 ± 465.9 | 77.7 ± 4.1 | 69.0 ± 0.8 | 27.7 ± 3.9 | 20.0 ± 5.1 | 0.4 ± 0.0 | 9.3 ± 0.6 | 1.7 ± 0.2 | 96.3 ± 7.5 | 4.1± 0.3 5.6 ± 0.4 1.4 ± 0.2 |
| 4 months age | 2 | Average ± SE | | 8.4 ± 0.4 | 5.8 ± 0.2 | 0.1 ± 0.0 | 32.5 ± 5.5 | 29.0 ± 2.0 | 36.0 ± 4.0 | 60.0 ± 16.0 | 310.5 ± 50.5 | 42.0 ± 3.0 | 809.5 ± 503.5 | 953.0 ± 167.0 | 80.0 ± 14.0 | 54.0 ± 1.0 | 21.5 ± 0.5 | 31.0 ± 8.0 | 0.4 ± 0.0 | 12.6 ± 0.5 | 1.7 ± 0.0 | 76.0 ± 7.0 | 4.2± 0.1 5.1 ± 0.2 1.7 ± 0.0 |
| 6 months age | 1 | Average | | 8.9 | 6.0 | 0.1 | 57.0 | 26.0 | 31.0 | 186.0 | 261.0 | 32.0 | 727.0 | 746.0 | 161.0 | 60.0 | 22.0 | 44.0 | 0.5 | 12.1 | 2.2 | 97.0 | 4.2 5.5 1.8 |

All data are expessed as mean±S.E.

S.E.: standard error

Day0: The day of beginning of treatment

1 week treatment: fed contaminated haylage for 1 week

1 month treatment: fed contaminated haylage for1 month

1 month age: 1 month after from stopped fed contaminated haylage

4 months age: 4 month after from stopped fed contaminated haylage

6 months age: 6 month after from stopped fed contaminated haylage

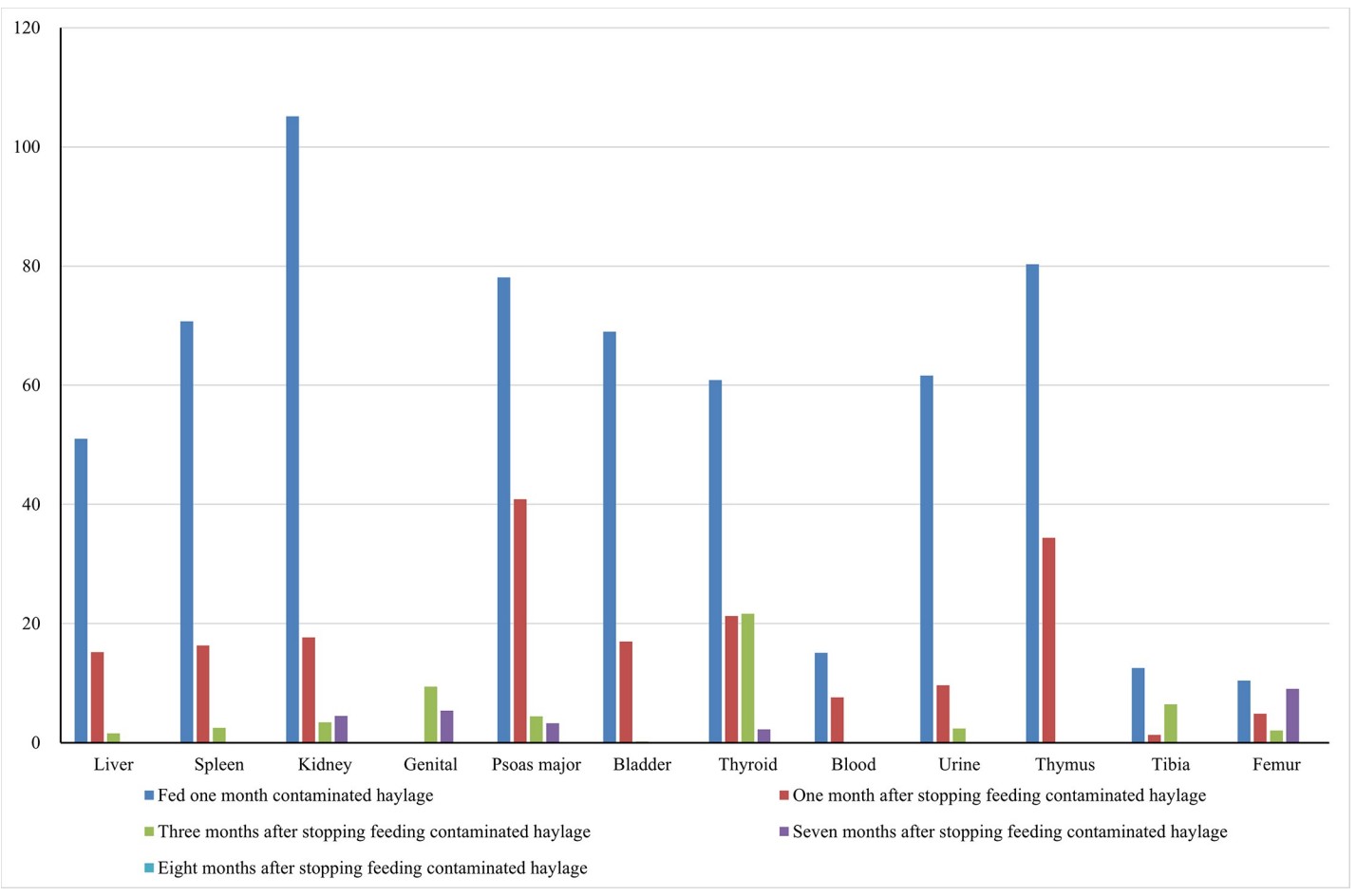

**Fig 1. Radiocesium level of body tissue and organs (Growing pig Bq/kg).** The time dependent radiocesium exclusion level changes of the different body tissues and organs (Bc/kg). The body tissues and organs were from the growing pigs fed with low level cesium contaminated haylage for one month.

were detected. The type of pasture has been shown to have a significant effect on cattle ALP-P III values, but not on other biochemical indices [9]. In the present study, pigs were fed haylage, although not within a pasture. ALP levels have been shown to vary widely with age and sex in children and adolescents [10]; therefore, the large variation observed in the control and treatment ALP levels in the present study may have been caused by factors other than radiocesium administration, especially considering the good health condition of the pigs.

Adult pigs had lower WBC counts in the treatment group than in the control group; however, these differences were not accompanied by differences in RBC count and PLT level, and no clinical symptoms were detected. Our biochemical data showed that the TP-P III and ALB-P levels decreased and the NH3-P II levels increased after the first week of treatment, perhaps due to mixing haylage with concentrated feed, which was confirmed to affect the amino acid balance. Amino acid imbalances can negatively affect protein synthesis and increase protein metabolism, resulting in a high blood ammonia concentration. However, no symptoms of these processes were observed in the juvenile pigs.

After 1 month of treatment, when radiocesium accumulation was expected to be highest, the radiocesium levels were high in all organ samples from juvenile pigs, but lower in blood and bone. Tibia and femur samples require longer extraction times [1]. We obtained no genital radioactivity data for juvenile pigs in the present study because the males were castrated at

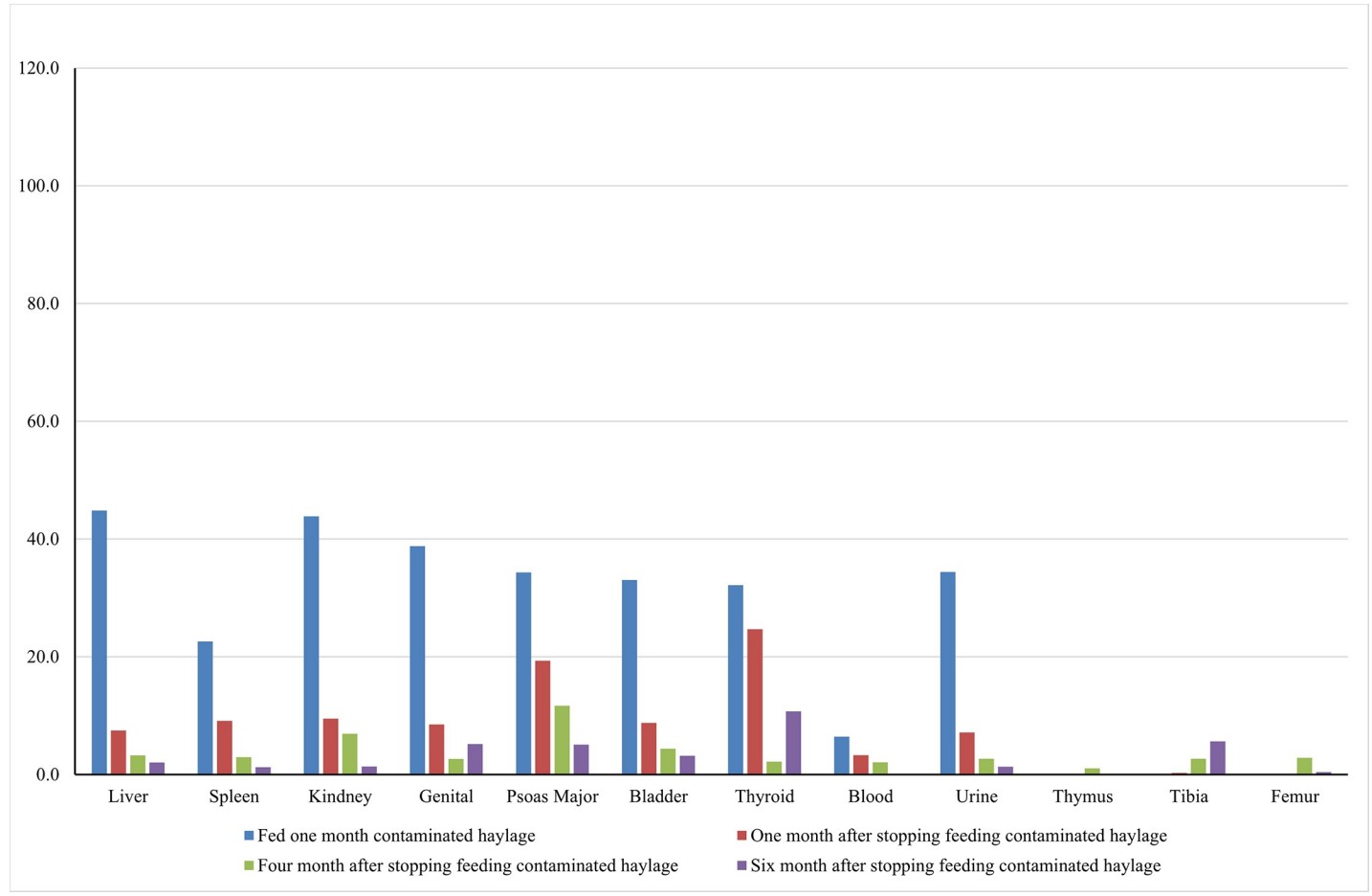

**Fig 2. Radiocesium level of body tissue and organs (Adult pig Bq/kg).** The time dependent radiocesium exclusion level changes of the different body tissues and organs (Bc/kg). The body tissues and organs were from the adult pigs fed with low level cesium contaminated haylage for one month.

**Table 7. Transfer coefficients of radionuclides (%).**

| Transfer Coefficients of Radionuclides from Fed to Livestock Products | Adult | Growing pigs |
|---|---|---|
| Liver | 6.9 | 7.9 |
| Spleen | 3.5 | 10.9 |
| Kidney | 6.8 | 16.2 |
| Genital | 6.0 | |
| Psoas Major | 5.3 | 12.1 |
| Bladder | 5.1 | 10.6 |
| Thyroid | 5.0 | 9.4 |
| Blood | 1.0 | 2.3 |
| Urine | 5.3 | 9.5 |
| Thymus | | 12.4 |
| Kibia | 0 | 1.9 |
| Femur | 0 | 1.6 |
| Average | 4.1 | 8.6 |

nursing and the female ovaries were too small. Radiocesium retention in all organs of the adult pigs was lower than that in the juvenile pigs. Stable equilibrium was reached quickly in similar-sized voles [4] and after 1 month of continuous radiocesium administration in wild and captive hispid cotton rats [11]. Therefore, we expected that 1 month of radiocesium treatment would be sufficient for stable equilibrium in the present study.

We also found that juvenile pigs had higher transfer coefficient rates than adult pigs, which suggests that age is an important factor in radiocesium retention. The radiocesium elimination time for both growth stages was 5–6 months or longer. Transfer coefficient rates have been found to be higher in lambs than in than adult sheep [12, 13]. These results are consistent with those of the present study; however, effective Cs retention was found to be lower in younger mice, and progressively higher in older mice [14, 15]. These inconsistent results may be due to differences in treatment duration since both older studies administered single-dose treatments.

The blood and bone samples had lower radiocesium levels than other organs in both adult and juvenile pigs. Radiocesium was not detected in adult bone samples 1 month after the end of treatment, when radiocesium accumulation was expected to be highest, but increased at 4 and 6 months post-treatment. Few studies have reported similar results, although Lei [4] found that the lowest radiocesium levels in *Microtus canicaudus* organ samples were detected in bone.

Get together the radiocesium transfer coefficient rates after 1 month of chronic oral treatment, showed higher radiocesium retention rates in the kidneys, liver, spleen, genitals, psoas major, bladder, thyroid, and urine than in the blood and bone (tibia and femur) of pigs at both growth stages. And the radiocesium retention levels were generally higher in juvenile pigs than in adult pigs, with the highest transfer coefficient ratio in the kidneys.

## Author Contributions

**Data curation:** Junyou Li.

**Formal analysis:** Chunxiang Piao, Junyou Li.

**Investigation:** Chunxiang Piao, Min Ma, James K. Chambers, Kazuyuki Uchida, Masanori Ikeda, Natsuko I. Kobayashi, Atsushi Hirose, Keitaro Tanoi, Masayoshi Kuwahara.

**Supervision:** Junyou Li.

**Writing – original draft:** Chunxiang Piao.

**Writing – review & editing:** Junyou Li.

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
