## [Decision Letter · Decision Letter 0]

23 Jul 2020

PONE-D-20-12149

Radiocesium transfer rates among pigs fed haylage contaminated with low levels of

cesium at two differentiation stages

PLOS ONE

Dear Dr. LI,

Thank you for submitting your manuscript to PLOS ONE. After careful consideration, we feel that it has merit but could be improved somewhat by incorporating the suggestions of the reviewers. Therefore, we invite you to submit a revised version of the manuscript that addresses the points raised during the review process. These are relatively and should be easy to address. So, congratulations!

We look forward to receiving your revised manuscript.

Kind regards,

Tim A. Mousseau

Academic Editor

PLOS ONE

Journal Requirements:

2.  Thank you for stating the following in the Competing Interests/Financial Disclosure * (delete as necessary) section:

"This study was funded by a Japan Society for the Promotion of Science KAKENHI Grant-in-Aid for Scientific Research (C) (no. JP25517004) and by the Livestock Promotion Agency of Japan Racing Association."

We note that you received funding from a commercial source: "Japan Racing Association"

3. Please include your tables as part of your main manuscript and remove the individual files. Please note that supplementary tables (should remain/ be uploaded) as separate "supporting information" files

Reviewers' comments:

Reviewer's Responses to Questions

**Comments to the Author**

1. Is the manuscript technically sound, and do the data support the conclusions?

Reviewer #1: Yes

Reviewer #2: Yes

2. Has the statistical analysis been performed appropriately and rigorously? 

Reviewer #1: Yes

Reviewer #2: Yes

3. Have the authors made all data underlying the findings in their manuscript fully available?

Reviewer #1: Yes

Reviewer #2: Yes

4. Is the manuscript presented in an intelligible fashion and written in standard English?

Reviewer #1: Yes

Reviewer #2: Yes

5. Review Comments to the Author

Reviewer #1: The authors studied the rate of radiocaesium transfer in pigs at different growth stages after feeding with low levels of caesium-contaminated haylage. In my opinion, this manuscript was well designed and providing valuable data. Just a few suggestions to improve the quality of manuscript.

1. The value and significance of this study should be proposed in the last sentence of the Abstract.

2. The first three paragraphs of Introduction needed to be properly integrated.

3. Although the authors described a series of indicators tested in this study in the last paragraph of Introduction, there was still a need to describe the meaning behind these indicators.

4. Why did the author select only four adult pigs? In addition, in the absence of a control group, how could the author ensure that the detected indicators are statistically significant?

5. Why the authors chose 0.7 kg of contaminated haylage to feed the pigs.

6. Why was the current point of time chosen to collect the blood sample?

7. The first paragraph of the Discussion seemed more appropriate in the Materials and Methods section.

8. A general conclusion should be added to the last paragraph of the Discussion.

9. All Tables should use a three-line table and, in addition, the number of biological duplicates should be indicated in the table notes.

10. The captions and order of Figure 1 and 2 were incorrect and needed to be corrected.

Reviewer #2: There is no explanation for many abbreviations in the table and figures.

Figure 1- what does stop treatment in the legend

Table 1 & 2- what are units for weight

Table 3 & 4- what are (T) and (C) in column 2.

Table 5- what is SD in column 1

6. PLOS authors have the option to publish the peer review history of their article (what does this mean?). If published, this will include your full peer review and any attached files.

Reviewer #1: No

Reviewer #2: **Yes: **Steve Ensley

---

## [Author Response · Author response to Decision Letter 0]

4 Aug 2020

I would like to thank the reviewers.

I am prepared the manuscript according to the PLOS ONE's style requirements and I believe that revised manuscript could be meet your requirement for publishing in journal PLOS ONE. The details answer to reviewer’s comments sent by the attached file.

---

## [Editor Report · Decision Letter 1]

7 Aug 2020

Radiocesium transfer rates among pigs fed haylage contaminated with low levels of

cesium at two differentiation stages

PONE-D-20-12149R1

Dear Dr. LI,

We’re pleased to inform you that your manuscript has been judged scientifically suitable for publication and will be formally accepted for publication once it meets all outstanding technical requirements.

Kind regards,

Tim A. Mousseau

Academic Editor

PLOS ONE
---

## [Editor Report · Acceptance letter]

13 Aug 2020

PONE-D-20-12149R1 

Radiocesium transfer rates among pigs fed haylage contaminated with low levels of cesium at two differentiation stages 

Dear Dr. LI:

I'm pleased to inform you that your manuscript has been deemed suitable for publication in PLOS ONE. Congratulations! Your manuscript is now with our production department. 

Kind regards, 

on behalf of

Dr. Tim A. Mousseau 

Academic Editor

PLOS ONE